# TRUNCPROOF: LL(1)-CONSTRAINED GENERATION IN LARGE LANGUAGE MODELS WITH MAXIMUM TOKEN LIMITATIONS

## ABSTRACT

The generation of machine-readable outputs using LLMs has attracted significant attention. However, existing approaches cannot strictly enforce the maximum number of tokens to be generated. To address this limitation, we propose TruncProof, a novel grammar-constrained generation method that enables LLMs to produce grammatically valid outputs while adhering to a predefined token limit. By leveraging the properties of LL(1) parsers, TruncProof efficiently estimates the minimum number of tokens required to complete a grammatically valid output at each decoding step. Experiments on the Text-to-JSON instruction task and Code generation task demonstrate that TruncProof successfully generates syntactically correct outputs even under strict token constraints. Furthermore, we show that TruncProof can be effectively combined with advanced decoding strategies, resulting in outputs that are not only grammatically valid but also semantically accurate. The source code will be made public upon acceptance.

## 1 INTRODUCTION

Recently, there has been a growing body of research on solving complex tasks by combining the code generation capabilities of large language models (LLMs) with external tools such as Python interpreters (Wang et al., 2024) and neuro-symbolic systems (Gupta & Kembhavi, 2023). For these applications to be reliable, LLMs must consistently produce well-formed, machine-readable outputs. However, most LLM tokenizers are designed for natural language, making it difficult to ensure grammatically valid outputs through fine-tuning or prompting alone. To address this robustness issue, several grammar-constrained generation (GCG) methods have been proposed (Scholak et al., 2021; Poesia et al., 2022; Beurer-Kellner et al., 2023; Lundberg et al., 2023; Willard & Louf, 2023; Gerganov et al., 2023; Beurer-Kellner et al., 2024; Ugare et al., 2024; Dong et al., 2025). Recent approaches typically rely on context-free grammar (CFG) parsers, which can express a wide range of machine-readable formats and programming languages.

While these methods can enforce complex grammatical constraints on LLM outputs, they have a critical limitation: *they cannot strictly enforce a maximum number of generated tokens*. In practical applications, imposing a token limit is essential to prevent infinite generation, control memory usage, and keep the output within the model's context window. However, because current constraint-based methods cannot dynamically estimate the number of tokens needed to complete a grammatically valid output, they terminate generation abruptly once the token limit is reached, often resulting in incomplete or grammatically invalid outputs. This issue is particularly problematic in agent-based applications, where autonomous agents are required to quickly exchange structured text without human intervention; such termination leads to parse errors that can subsequently disrupt downstream processes.

To address this truncation issue, we propose a novel GCG guardrail that enables LLMs to generate grammatically correct outputs while adhering to a specified maximum number of tokens. This requires estimating, at each decoding step, the minimum number of tokens needed to complete a grammatically valid output. We address this challenge by leveraging the properties of LL(1) parsers (Aho & Ullman, 1972), which accept a diverse subset of CFGs (Parr & Fisher, 2011). Unlike the CFG parsers employed in existing methods (*e.g.*, LR(*) parsers), LL(1) parsers can determine gram-

matically permissible continuations given a partially generated sequence. This property allows us to compute the shortest valid token sequence required to complete the output at each step. With this information, we construct constraint masks to prevent the selection of tokens that would violate the grammar or token limit. We formally describe our approach and provide theoretical guarantees (see § 4 and § B.5, B.6 and B.7 of our supplementary material).

Our proposed method, called TruncProof hereafter, has a form of logit modifier. Therefore, it is compatible with a wide range of tokenizers, language models, other logit modifiers and various decoding strategies. We evaluate TruncProof on the Text-to-JSON instruction task (NousResearch, 2024) and Code generation task. Experimental results show that TruncProof enables LLMs (*e.g.*, Google, 2024, Touvron et al., 2023) to produce grammatically valid JSON outputs, even under strict token budget constraints, whereas existing methods almost fail to do so. Furthermore, by incorporating advanced decoding strategies such as Beam Search and Monte Carlo Tree Search, TruncProof significantly enhances the semantic robustness of the JSON and C outputs while preserving grammatical validity, whereas existing methods fail to achieve this balance.

## 2 BACKGROUND

To enhance self-containment, we first introduce the foundation of Grammar-Constrained Generation in §2.1. We then provide an overview of Context-Free Grammars in §2.2, followed by implementations of its parsers in §2.3. Throughout this paper, we denote the finite set of characters that can be generated by an LLM as $\Sigma$, and the set of all finite-length strings over $\Sigma$ as $\Sigma^*$ [1]. The empty string is denoted by $\epsilon$, and the concatenation of two strings $w$ and $v$ is represented as $(w.v)$.

### 2.1 GRAMMER-CONSTRAINED GENERATION (GCG)

Modern LLMs generate output tokens from a vocabulary $\mathcal{V}$ in an auto-regressive manner: At each generation step $i$, the model takes the current partial output $t_{<i} = t_1.\cdots.t_{i-1} \in \mathcal{V}^*$ and predicts the probability distribution of the $i$-th token $P(t_i \mid t_{<i})$. In Grammar-Constrained Generation (GCG), *constraint functions* evaluate the grammatical validity of each candidate token $t_i$ at every step. Specifically, given a string $t_{<i}$, the constraint function uses a *parser* to check whether there exists a string $w$ that extends the candidate token into a grammatically valid sentence, and returns the result in the form of a *constraint mask* $\mathbf{m}$. Formally, the element of $\mathbf{m}$ for a next token candidate $t$, $m_t$, is defined as follows:

$$m_t = true \;\Rightarrow\; \exists w \in \mathcal{V}^* \text{ s.t. } (t_{<i}.t.w) \in L(G), \tag{1}$$

where $G$ is a grammar and $L(G)$ is the *language* defined as the set of strings accepted by $G$. Tokens deemed grammatically invalid are re-assigned zero probability by element-wise multiplication between the probability distribution and the constraint mask *i.e.*, $P(t_i \mid t_{<i}) \odot \mathbf{m}$. Note that this modification is applied prior to selecting the next token for generation. Consequently, from an algorithmic perspective, any GCG method, including our proposed TruncProof, can be combined with various decoding strategies. Details are provided in §4.2.

### 2.2 CONTEXT-FREE GRAMMAR (CFG)

Context-Free Grammar (CFG) has been used to define a variety of machine-readable formats. CFG is characterized by a four-tuple $(\mathcal{N}, \Sigma_T, R, S)$: a finite set of the *nonterminal* symbols that does not appear in the language $\mathcal{N}$, a finite set of the *terminal* symbols as the alphabet in the language $\Sigma_T$, a finite relation which represents derivation rules that rewrite a single nonterminal to the terminal or nonterminal symbols with 0 or more length $R \subset \mathcal{N} \times (\mathcal{N} \cup \Sigma_T)^*$, and the start symbol $S \in \mathcal{N}$. Using this expression, we can define the language $L(G)$ as the set of the terminal sequences. Any terminal sequence $\sigma \in \Sigma_T^*$ in the language can be generated by repeated derivations (denoted as $\rightarrow^*$) from the start symbol. CFG parsers must construct a derivation process that generates the string from the start symbol to determine whether the string belongs to the language. Notice that these processes can be visualized as derivation trees, with the start symbol at the root and terminal symbols at the leaves. An example of a CFG and its derivation process is provided in §B.4 of our supplementary material.

---

[1] For example, when $\Sigma = \{a, b, c\}$, $\Sigma^* = \{\epsilon, a, b, c, aa, ab, ac, ba, \cdots\}$.

Usually, to prevent grammars being too complicated, terminal symbols in CFG are defined as *Regular Expression (Regex)* instead of characters (Shinan, 2017) and the parsers preprocess the input string to identify the equivalent terminal sequence. Regex can be parsed by using *Deterministic Finite Automaton (DFA)*, which characterized by a five-tuple $(Q, \Sigma, \delta, q_0, F)$: a finite set of states $Q$, a finite set of recognizable characters $\Sigma$, a transition function that determines the next state based on a current state and a captured character $\delta : Q \times \Sigma \to Q$, the initial state $q_0 \in Q$, and a set of accepting states $F \subseteq Q$. DFA starts from the initial state and accepts the input if and only if its state transitions to an accepting state by processing each character one by one.

## 2.3 IMPLEMENTATIONS OF CFG PARSERS

There are two primary approaches to implement CFG parsers (Aho & Ullman, 1972): **The bottom-up approach**, such as LR(*) parsers, which identifies the derivation tree from the bottom (*i.e.*, from the leaf nodes), and **the top-down approach**, such as LL(*) parsers, which constructs the derivation tree from its top (*i.e.*, from the root). Their distinction is reflected in the structure of the partially constructed derivation tree when they process incomplete input, as illustrated in Figure 1. Contrary to bottom-up parsers, top-down parsers can easily enumerate possible continuations of the current input by applying arbitrary derivations from the unexpanded nonterminals. To leverage this advantage and ensure that the content of the derivation tree is deterministically fixed at each generation step, our TruncProof employs LL(1), a top-down parser that permits only single-terminal lookahead without allowing backtracking (reconstruction of the derivation tree). Note that the LL(1) grammars (*i.e.*, grammars supported by LL(1) parsers) form a strict subset of CFGs. Although LL(1) does not support all Context-Free languages, it still supports sufficiently expressive grammars with unlimited enumeration and deeply nested structures such as JSON, which is the de-facto standard machine-readable format in practical systems (OpenAI; Anthropic; Google). A formal definition of LL(1) grammar based on Lewis & Stearns (1968) is described in Appendix B.1.

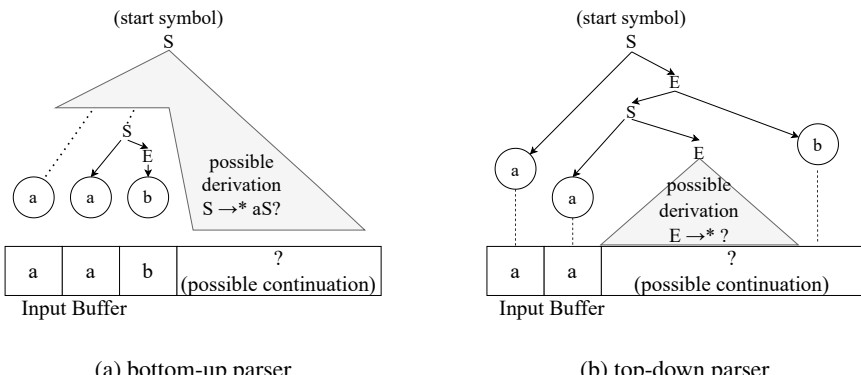

(a) bottom-up parser       (b) top-down parser

Figure 1: Examples of partially constructed derivation trees generated by two different parsers.

## 3 RELATED WORKS

Several GCG methods have been proposed in recent years, most of which can be classified based on the type of grammar they support. For example, PICARD (Scholak et al., 2021) is designed for SQL, where it generates multiple candidates simultaneously and checks the parsability of each. LMQL (Beurer-Kellner et al., 2023) allows user-defined grammars based on Regex through a custom specification language. Outlines (Willard & Louf, 2023) improves the efficiency of Regex-based generation by precomputing valid token sets for each DFA state. Although Outlines also supports CFGs, it is usually slow since it repeats sampling and validation of candidates until a grammatically valid token is found. Recently, research has widely been conducted to further optimize precomputation or runtime processing within the scope of CFGs: DOMINO (Beurer-Kellner et al., 2024) and SynCode (Ugare et al., 2024) integrate optimized Regex validation with the CFG parsers that enumerate acceptable terminal sequences. XGrammar (Dong et al., 2025) introduces a variant of CFG parser that operates on characters rather than terminals, thereby reducing the overhead associated

with terminal processing. LLGuidance (Moskal et al., 2025) adopts trie trees to handle LLM tokens with low-level optimization to reduce the overhead in runtime. GreatGramma (Park et al., 2025) aggregates all terminal definitions and the LLM vocabulary into a single Finit State Transducer that processes input token by token, which largely reduces the preprocessing cost.

While the above methods can impose sufficiently complex grammatical constraints on LLMs, they share a common limitation: they cannot ensure that generation halts within a specified number of tokens. IterGen (Ugare et al., 2025) can address this problem by repeatedly regenerating outputs until a desired result is obtained. However, it does not guarantee that a grammatically correct output will be found within a reasonable number of iterations.

We also note that the literature includes methods that extend beyond CFG-based constraints. Mündler et al. (2025) and Li et al. (2025) propose a code generation framework that imposes richer constraints than CFGs, aiming to avoid any errors during compilation or execution. While this direction is promising, these methods abandon constraint mask generation and instead rely on inefficient candidate sampling, similar to Outlines, which is especially disadvantageous when combined with advanced decoding strategies. Geng et al. (2023) introduces token-level grammars that directly provide next valid tokens and supports more flexible grammars than CFGs. However, this token-level approach potentially results in worse perplexity, since it prohibits to generate the same string consisting of natural token combinations.

## 4 TRUNCPROOF

Let a grammar $G$ be specified in the form of an LL(1) grammar $(\mathcal{N}, \Sigma_T, R, S)$. We assume that each terminal symbol in $\Sigma_T$ is defined by a Regex; For each terminal, there exists a corresponding DFA $\mathcal{M}_a := (Q_a, \Sigma, \delta_a, q_{a0}, F_a)$ that accepts the strings defined by the Regex. Given a grammatically valid partial output $t_{<i}$, our TruncProof serves as a constraint function that returns the binary mask $\mathbf{m}$, where each entry $m_t$ represents the grammatical validity of a token $t \in \mathcal{V}$ within the pre-defined token limit $N_{max}$. By extending Equation 1, $m_t$ is formally defined as follows:

$$m_t = true \implies \exists w \in \mathcal{V}^* \text{ s.t. } ((t_{<i}.t.w) \in L(G) \text{ and } |t_{<i}.t.w| \le N_{max}). \tag{2}$$

This mask can be used to filter out tokens that would result in either (1) a grammatically invalid continuation or (2) an output exceeding $N_{max}$.

In § 4, we describe the details of TruncProof, which returns the mask $\mathbf{m}$. Note that this mask ensures grammatical validity but does not fully account for semantic correctness. To produce outputs that are both grammatically valid and semantically coherent, we extend TruncProof with advanced decoding strategies, as detailed in § 4.2.

### 4.1 DETAILS OF TRUNCPROOF

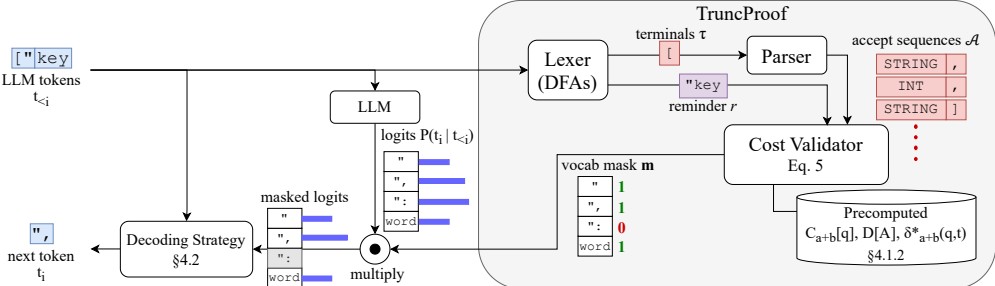

Figure 2: Overview of TruncProof. For $i$-th generation step, Lexer parses the intermediate LLM tokens generated by the LLM into the terminals $\tau$ and the reminder $r$, Parser collects all possible terminal sequences (called accept sequences $\mathcal{A}$) whose length is at most two, and Cost Validator constructs the vocabulary mask $\mathbf{m}$ by validating the future cost for each candidate token based on the precomputed cache.

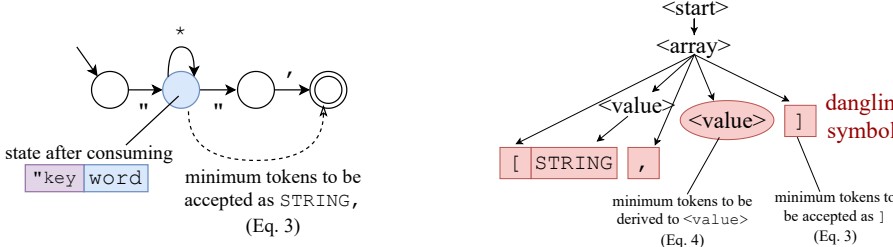

(a) Counting future tokens that are accepted by the DFA for STRING followed by COMMA. (b) Counting future tokens to finish output after accepting ["keyword",

Figure 3: The examples of counting the future tokens in Cost Validator illustrated in Figure 2.

Figure 2 illustrates the overall structure of TruncProof. In runtime, the following steps are executed iteratively within the generation loop: (i) Given the intermediate output generated by the LLM, Lexer that handles Regex and Parser that handles LL(1) grammar incrementally parse the newly generated token based on the terminal sequence obtained in the previous iteration. (ii) Cost Validator estimates the number of tokens needed in the future assuming a next token (as illustrated in Figure 3), and verifies whether the generated output remains grammatically valid under the specified token budget.

To efficiently operate Cost Validator, we precompute the estimation of the shortest token lengths for realizing any terminal and nonterminal defined by the given LL(1) grammar. In the following sections we describe the behavior in the runtime phase and the things to be prepared in the precomputation phase.

### 4.1.1 RUNTIME PHASE

As shown in Figure 2, we first divide the intermediate input $t_{<i}$ into the terminal sequence $\tau \in \Sigma_T^*$ and the reminder[2] $r \in \Sigma^*$ by using the DFAs, then partially parse $\tau$ to identify the derivation tree by using the LL(1) parser. This process can be executed incrementally by using the results in the previous iteration. Next, we enumerate the terminal sequences with a length of at most two *i.e.*, $a, b \in \Sigma_T$, that can be given to the current parser in this generation step. We hereafter call the set of the sequences as *accept sequence* $\mathcal{A} \subseteq \Sigma_T \cup \Sigma_T^2$. The reason why we take two-length terminals in consideration is because this extension allows us to better exploit the generative capabilities of the LLM[3] while the relaxed constraint still ensures the condition defined in Equation 2. After that, we calculate the two types of cost to complete the generation: the number of tokens to complete the reminder as terminals $(a, b)$ (as illustrated in Figure 3a), and the further cost $d_{cost}(\tau.a.b)$ to complete the whole string after $a$ and $b$ are accepted by the parser (as illustrated in Figure 3b). The former cost can be estimated as the minimum number of tokens required to transition from each state $q$ in the corresponding DFA $M_{a+b}$ to an accepting state, which is formulated as follows:

$$C_{a+b}[q] := \begin{cases} \min_{w \in \mathcal{V}^*} |w| \text{ subject to } \delta_{a+b}^*(q, w) \in F_{a+b} & (\text{if } \exists w \text{ s.t. } \delta_{a+b}^*(q, w) \in F_{a+b}) \\ \infty & (\text{otherwise}), \end{cases} \quad (3)$$

where $\delta_a^*$ is an iterated transition function *i.e.*, $\delta_a^*(q, x_1. \cdots .x_n) = \delta_a(\cdots \delta_a(q, x_1) \cdots, x_n)$. If there is no token sequence $w$ which can reach to any accepting state from $q$, $C_{a+b}[q]$ is set to infinity. This ensures that grammatically invalid tokens are automatically excluded due to their infinity cost. The latter cost $d_{cost}(\tau.a.b)$ is computed as the sum of the minimum number of tokens to consume the terminals and nonterminals that remains unresolved by the LL(1) parser (the dangling symbols illustrated in Figure 3b). To compute it, we need the approximate shortest token length

---

[2] User-defined terminal symbols may not align exactly with LLM tokens. In such cases, some suffixes of the output remain unprocessed as reminders.

[3] For instance, in the case of JSON, by precomputing the constraint mask for the concatenation of a left brace and a string, we can treat a token such as {" as a valid starting sequence of a JSON object. This allows the model to generate more natural and compact outputs while still adhering to the grammatical constraints.

$D[A]$ derivable from each nonterminal $A \in \mathcal{N}$, by the following equation:

$$D[A] := \min_{\sigma \in \Sigma_T^*} \sum_{i=1}^{|\sigma|} C_{\sigma_i}[q_{\sigma_i 0}] \text{ subject to } A \to^* \sigma, \tag{4}$$

where $\sigma_i$ denotes the $i$-th terminal symbol in the sequence $\sigma$. In summary, the entry of the constraint mask $\mathbf{m}^{(a,b)}$ for a token $t$, *i.e.*, $m_t^{(a,b)}$, is computed as follows:

$m_t^{(a,b)} := true$ iff.

$$\underset{\substack{\text{(consumed} \\ \text{tokens)}}}{i} + \underset{\substack{\text{(future tokens} \\ \text{that DFA accepts)}}}{C_{a+b}[\delta_{a+b}^*(q_{a+b0}, r.t)]} + \underset{\substack{\text{(future tokens} \\ \text{to finish output)}}}{d_{cost}(\tau.a.b)} < N_{max}, \tag{5}$$

where $i$ is the number of generated tokens. Once the simulation of the parser and the calculation of the future cost are performed, the constraint mask $\mathbf{m}$ can be obtained by taking the element-wise union of the masks $\mathbf{m}^{(a,b)}$ for each $(a, b) \in \mathcal{A}$. Since each valid entry corresponds an actual sequence of tokens, it guarantees the result that adheres to the grammar and token limit. For the proof of this guarantee, refer to §B.7 in our supplementary material.

**Time Complexity Analysis.** At each iteration of the generation loop, the computational bottle-neck is the simulation of the LL(1) parser to calculate $d_{cost}(\tau.a.b)$ for each $(a, b) \in \mathcal{A}$. It takes $O(|\Sigma_T|^2(T_G + |\Gamma|))$, where $T_G$ is the cost to feed one terminal to the LL(1) parser and $|\Gamma|$ is the number of dangling symbols in the derivation tree, which tends to be proportional to the nesting depth of the output code. In practice, $|\Sigma_T|$ is not so large; JSON has about 15 terminals and Ugare et al. (2024) reports that Python has 94. Calculation of $\delta_{a+b}(q_{a+b0}, r.t)$ can be accelerated by pre-computing the mapping $\delta_{a+b}^*(q, t)$ for each terminal, DFA state, and LLM token. At runtime, we calculate the state $q' = \delta_{a+b}(q_{a+b0}, r)$ and lookup the precomputed state $\delta_{a+b}^*(q', t)$ for each termi-nal sequence $(a, b)$ and token $t$. This lookup operation can be parallelized into a vector computation across the entire $\mathcal{V}$. Mask generation is processed by at most $|\Sigma_T|^2$ times of element-wise Boolean and arithmetic operations on the vector of length $|\mathcal{V}|$, which also can be parallelized. Notice that this cost is usually smaller than the brute force method that searches the shortest terminal sequence by simulating the parser; The cost is $O(|\Sigma_T|^D T_G)$, where $D$ is the minimum number of terminals in continuation, and $D$ tends to be proportional to the nesting depth of generated sentences.

### 4.1.2 PRECOMPUTATION PHASE

In this phase, we precompute the necessary values required for efficiently calculating Equation 5. First we calculate $C_a[q]$ provided in Equation 3 for each terminal $a \in \Sigma_T$ and $C_{a+b}[q]$ for each two-length terminals $(a, b)$. To compute them, we use Dijkstra's algorithm, treating DFA states as nodes, transitions as edges, and token lengths as edge costs. The pseudo-code is provided in Algorithm 1 of Appendix B.5. Next, we estimate $D[A]$ provided in Equation 4. The computation of $D[A]$ is also based on Dijkstra's algorithm, where possible derivation states are treated as nodes and derivation steps as edges. The corresponding pseudo-code is Algorithm 2 in Appendix B.5. Although the underlying search graph may be infinitely large in theory, our algorithm is guaranteed to terminate whenever the nonterminal $A$ can derive at least one terminal sequence. This is ensured by the property of LL(1) grammars, which prohibits infinitely recursive derivations without increasing the number of leading terminals. We present the formal proof of this termination in Appendix B.6. Finally, we precompute the mapping $\delta_{a+b}^*(q, t)$ for each terminal, DFA state, and LLM token. This is used to efficiently retrieve the DFA state in consuming a reminder and a LLM token illustrated in Figure 3a.

**Space Complexity Analysis.** The amount of memory for precomputation is the sum of the mem-ory $O(|\Sigma_T|^2|Q|)$ for $C_a[q]$, $O(|\mathcal{N}|)$ for $D[A]$, and $O(|\Sigma_T|^2|\mathcal{V}||Q|)$ for precomputing mapping $\delta_{a+b}^*(q, t)$, where $|Q|$ is the average size of the DFA states. Note that the mapping $\delta_{a+b}^*(q, t)$ is sparse because most tokens lead DFAs to a dead state.

### 4.2 COMBINING TRUNCPROOF WITH DECODING STRATEGIES

TruncProof can be seamlessly integrated with various decoding strategies. In this work we consider the following three decoding methods: (1) **Greedy decoding (Greedy)** is the default strategy in most text-generation libraries. It takes the token with the best likelihood $P(t \mid t_{<i})$ in each iteration

of the text generation. (2) **Beam Search (BS)** maintains $b$ best candidates in each iteration and re-selects the $b$ best sequences among the possible continuations. Scholak et al. (2021) adopts BS with their constraint method to improve the accuracy of the generation. Although BS takes diverse candidates into account and obtains better contents than the greedy strategy, it remains difficult to completely avoid future token shortages. (3) **Monte Carlo Tree Search (MCTS)** is known to be effective for this type of issue where the selections in beginning have a large effect but their precise value is evaluated in the ending phase. MCTS originally aims to find the best move in two-person games (Coulom (2006)), but there are some studies for LLM-based text generation (Leblond et al. (2021); Chaffin et al. (2022); Loula et al. (2025)). In each generation step $i$, MCTS constructs the search tree whose nodes are possible continuations $t_{<i+k}$ and edges are the selectable next tokens. MCTS repeats the following stages to grow the search tree: Selection, Expansion, Simulation, and Backup. In Selection, we traverse the tree up to a leaf based on the following evaluation function introduced by Silver et al. (2017) that utilizes the likelihood of sequences as a prior:

$$F(t_{<i}, t) := Q(t_{<i}, t) + c_{puct} P'_\tau(t \mid t_{<i}) \frac{\sqrt{\sum_u N(t_{<i}, u)}}{1 + N(t_{<i}, t)}, \tag{6}$$

where $Q(t_{<i}, t)$ is the maximum value observed among the continuations of $t_{<i}.t$, $P'_\tau$ is the likelihood modified by the constraint mask and normalized by softmax with temperature $\tau$, $N(t_{<i}, t)$ is the number of investigations beyond $t_{<i}.t$, and $c_{puct}$ is the hyperparameter that balances exploration and exploitation. In Expansion, we expand the tree to investigate more deeply beyond the leaf which we arrived at. In Simulation, we apply greedy decoding from the leaf until the end of generation and evaluate the value of the result text $v(t_{<n})$ as the geometric mean of the unmodified likelihood provided directly by the LLM, which is known as the inverse of the perplexity. In Backup, we tell the evaluated value $v$ to the ancestors and update their observed values $Q(t_{<i}, t)$. After some repetitions, we decide the next token $t$ with highest $Q(t_{<i}, t)$.

## 5 EXPERIMENTS AND DISCUSSION

We conduct the experiments on LL(1) grammars, JSON and a subset of C. Note that, in our experiments we do not consider Python, Go and SQL, which have been evaluated by Ugare et al. (2024), because they cannot be fully expressed using LL(1) grammars.

### 5.1 EXPERIMENTAL SETTING

**Quantitative Analysis on Text-to-JSON Instruction.** To evaluate TruncProof, we conduct experiments on the JSON-Mode-Eval dataset (NousResearch, 2024), which comprises 100 text-to-JSON tasks. In this instruction-following task, the goal is to generate syntactically and semantically valid JSON outputs given a natural language prompt (*cf*, Appendix B.2). In Ugare et al. (2024), the maximum token limit is fixed at 400, which is approximately six times the average length of the ground truth. To assess performance under stricter constraints, we define a more challenging configuration, where the maximum token length is dynamically set to $\lfloor L_i^{\text{GT}} \times e \rfloor$ for each instance $i$, with $L_i^{\text{GT}}$ denoting the token length of the ground truth and $e$ an expansion ratio. Unless otherwise specified, we set $e = 1.1$ when comparing TruncProof with other methods. For completeness, we conduct experiments under different token-limit settings, including the configuration used by Ugare et al. (2024), as well as various values of $e$. We also demonstrate the superiority of TruncProof over prompt engineering. The corresponding results are presented in Appendix B.8, B.10 and B.11 of the supplementary material, respectively.

As evaluation metrics, we use the following: (1) the percentage of outputs that are grammatically correct, denoted as *Syntax*; (2) the percentage of outputs that adhere to the schema specified in the prompt, referred to as *Schema*; and (3) the percentage of outputs that are parsed into JSON objects identical to the ground truth, termed *Exact-match*. The last Exact-match metric is newly introduced in this work to specifically assess the semantic validity of the generated JSON outputs.

Notice that the JSON grammar used in Ugare et al. (2024) does not fully comply with the official JSON standard, RFC 8259[4]. To ensure a practical and standards-compliant evaluation, we apply

---

[4]For example, numbers with a trailing decimal point such as `100.` are permitted by the grammar in Ugare et al. (2024), but are considered invalid under RFC 8259.

an RFC 8259-compliant JSON grammar (shown in Appendix B.3) to all constraint methods when assessing their performance.

**Qualitative Analysis on Code Generation.** In the experiments using JSON-Mode-Eval, we measure accuracy by checking whether keys and values in generated JSONs match exactly. Therefore, shorter JSON that maintains semantic meaning would be the one whose whitespace is reduced. To demonstrate how TruncProof with advanced decoding strategies can significantly alter content while preserving the semantics, we define the Code generation task to generate C functions that sums up 1 to $N$ using a limited C grammar adopted by Gerganov et al. (2023) with strict token limits. Under this setting, we observe the results of TruncProof and a prior work SynCode (Ugare et al., 2024).

**Environment.** We used 1x H200 GPU to produce all the results. Beam Search (BS) is performed with 10 beams while Monte Carlo Tree Search (MCTS) is performed with the following hyperparameters: $c_{puct} = 5, \tau = 2$, 20 trials for each generation step. We precompute the shortest token lengths for all terminals and nonterminals described in §4 before the experiments. It takes about 1 minute for the JSON grammar, and 5 minutes for the subset of C grammar.

Table 1: Accuracy and generation speed of JSON-mode-eval with $e = 1.1$. Time (ms) denotes the time of generating one token in milliseconds, and the value in parenthesis denotes the overhead of constrained generation, which is calculated by comparing with "No constraint". †XGrammar uses its builtin JSON grammar because its grammar format (EBNF) is incompatible with others (Lark).

| Model | Method | Decoding | Accuracy (%) | | | Time (ms) | |
|---|---|---|---|---|---|---|---|
| | | | Syntax | Schema | Exact-match | | |
| Gemma2-2B | No constraint | Greedy | 1 | 1 | 0 | 21.8 | |
| | Outlines (Willard & Louf, 2023) | Greedy | 36 | 33 | 22 | 458.7 | (+436.9) |
| | | BS | 4 | 4 | 2 | 4347.8 | (+4326.0) |
| | SynCode (Ugare et al., 2024) | Greedy | 4 | 3 | 0 | 23.5 | (+1.7) |
| | | BS | 1 | 1 | 0 | 54.0 | (+32.2) |
| | | MCTS | 4 | 4 | 0 | 438.6 | (+416.8) |
| | XGrammar † (Dong et al., 2025) | Greedy | 5 | 5 | 3 | **22.1** | **(+0.3)** |
| | | BS | 1 | 1 | 0 | 34.3 | (+12.5) |
| | | MCTS | 5 | 5 | 2 | 293.3 | (+271.5) |
| | Ours | Greedy | **100** | 62 | 21 | 25.7 | (+3.9) |
| | | BS | **100** | 85 | 37 | 60.8 | (+39.0) |
| | | MCTS | **100** | **86** | **58** | 518.1 | (+496.3) |
| Llama2-7B-Chat-HF | No constraint | Greedy | 2 | 2 | 0 | 17.6 | |
| | Outlines (Willard & Louf, 2023) | Greedy | 18 | 13 | 4 | 72.2 | (+54.6) |
| | | BS | 10 | 8 | 4 | 598.8 | (+581.2) |
| | SynCode (Ugare et al., 2024) | Greedy | 11 | 10 | 4 | 18.4 | (+0.8) |
| | | BS | 6 | 6 | 4 | 58.7 | (+41.1) |
| | | MCTS | 8 | 8 | 4 | 183.5 | (+165.9) |
| | XGrammar † (Dong et al., 2025) | Greedy | 11 | 9 | 2 | **18.3** | **(+0.7)** |
| | | BS | 5 | 3 | 2 | 32.5 | (+14.9) |
| | | MCTS | 9 | 8 | 3 | 175.1 | (+157.5) |
| | Ours | Greedy | **100** | 51 | 2 | 19.0 | (+1.4) |
| | | BS | **100** | 67 | 29 | 37.0 | (+19.4) |
| | | MCTS | **100** | **70** | **41** | 209.2 | (+191.6) |

## 5.2 RESULTS

Table 1 presents the results of five approaches: the baseline without any GCG method (denoted as No constraint), Outlines (Willard & Louf, 2023), SynCode (Ugare et al., 2024), XGrammar (Dong et al., 2025), and our proposed method, TruncProof. For the No constraint baseline, we adopt Greedy decoding. All constraint methods except Outlines are evaluated with Greedy, BS, and MCTS. Note that BS and MCTS are implemented by ourselves, as they are not provided by the original authors. Following prior work (Ugare et al., 2024), we use Gemma2-2B (Google, 2024) and Llama2-7B-Chat-HF (Touvron et al., 2023) as the underlying language models.

**Syntax Robustness.** As expected, under this challenging setting, most outputs generated by the baseline methods are grammatically invalid, with their Syntax accuracies ranging from only 1% to

36%. This failure occurs mainly because LLMs include excessive whitespace in JSON for readability and thereby waste LLM tokens. In contrast, TruncProof consistently produces grammatically valid outputs across all decoding strategies and backend LLMs, achieving perfect Syntax accuracy *i.e.*, 100%. These results clearly demonstrate the effectiveness of our approach in maintaining grammatical correctness under strict token constraints.

**Semantics Robustness.** Table 1 also shows that when using simple decoding strategies such as Greedy, the Exact-match accuracies of TruncProof remain relatively low (2%–21%) although about half (51%-62%) of the cases are faithful to the schema. We emphasize that this outcome is expected; TruncProof only cares about the grammar and the number of tokens, but it does not fully account for the semantic correctness of its outputs. Also as shown in the same table, these scores improve significantly when more advanced decoding strategies are employed. In particular, using BS raises the Exact-match accuracies to 29%–37%, and further improvements are observed with MCTS, reaching 41%–58%, all while preserving perfect grammatical correctness. These results highlight the compatibility of TruncProof with various decoding strategies and its ability to enhance semantic quality without compromising syntactic validity.

Also note that such compatibility with various decoding strategies is not necessarily supported by existing methods; As shown in Table 1, prior works with BS performs worse than Greedy. This may be attributed to the presence of many high-likelihood candidates that are grammatically invalid. To validate this hypothesis, in Figure 4, we visualize the perplexity of outputs under token shortage (labeled "Reached limit") for both SynCode (Ugare et al., 2024) and our TruncProof. As shown, when generation is constrained by SynCode, the perplexity of truncated outputs is worse than that of exact-match outputs (*i.e.*, successful generations), yet still better than the perplexity of the ground truth (see Figure 4a). This indicates that simply optimizing for likelihood under SynCode may lead to grammatically incorrect outputs due to local optima. In contrast, when our method reaches the token limit and generates unnatural outputs, the perplexity becomes worse than that of the ground truth, suggesting that TruncProof avoids such invalid local optima by preserving grammatical correctness throughout generation (see Figure 4b).

The result of the Code generation is demonstrated in Figure 5. We find that TruncProof with MCTS generates the simpler algorithm whereas SynCode (Ugare et al., 2024) with MCTS fails to find a better solution than Greedy. Notice that the perplexities exhibit the same trend as in Figure 4; Truncated codes found by SynCode are judged more "natural" by LLMs than the shorter, correct code produced by TruncProof. These findings also indicate that prior methods do not consistently benefit from advanced decoding strategies, whereas TruncProof does.

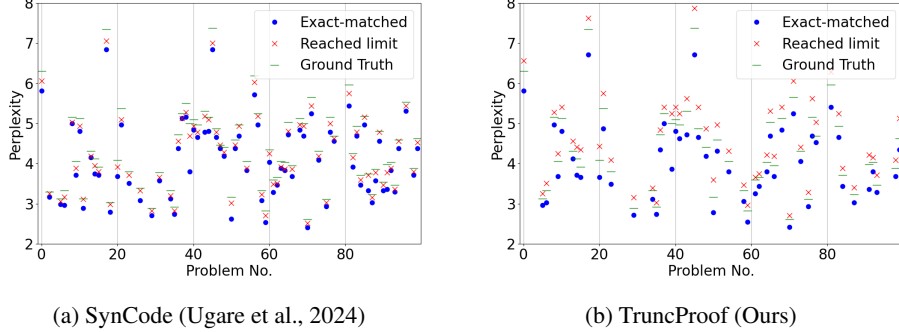

(a) SynCode (Ugare et al., 2024)  (b) TruncProof (Ours)

Figure 4: The perplexities provided by Gemma2-2B on JSON-Mode-Eval. Exact-matched indicates the output whose keys and values are correct under the relaxed token limit. Reached limit indicates the output which is truncated in (a) or incorrect in (b) due to the strict token limit. Refer to §5.2 for more details.

## 6 LIMITATIONS

As demonstrated in § 5.2, TruncProof is capable of generating both syntactically and semantically valid outputs under strict token budget constraints, particularly when paired with advanced decoding strategies. However, these strategies can slow down the generation process (*e.g.*, BS is 2.0-2.4x

| No Constraint (PPL 17.125) | TruncProof (PPL 70.0) Incorrect | TruncProof + MCTS (PPL 63.75) Correct |
|---|---|---|

```c
```c
int sum_to_n(int n) {
  int sum = 0;
  for (int i = 1; i <= n; i++) {
    sum += i;
  }
  return sum;
}
```...
```

```c
int sumToN(int N) {
  int sum = 0;
  for (int i = 1; i <= N; i = i +1) { } }
```

```c
int sum_to_n(int n) {   return n * (n + 1) / 2; }
```

SynCode (PPL 50.5) Syntax error

SynCode + MCTS (PPL 50.5) Syntax error

```c
int sumToN(int N) {
  int sum = 0;
  for (int i = 1; i <= N; i = i + 1)
```

```c
int sumToN(int N) {
  int sum = 0;
  for (int i = 1; i <= N; i = i + 1)
```

Figure 5: Responses of Gemma2-2B and their perplexity (PPL) for the prompt *"Write a C function that sums up 1 to N. Only output the code without codeblock quotations."* Without grammar constraint, the response has 58 tokens. When we apply SynCode or our TruncProof, we set the token limit to 40. The applied grammar is described in Appendix B.9.

slower and MCTS is 11.0-20.2x slower than Greedy). Although successful integration with the strategies is unattainable by other methods, the associated overheads may pose a practical limitation, especially in latency-critical applications.

Another potential limitation of TruncProof lies in its reliance on LL(1) parsing, which cannot support all CFGs. For example, in Python 3.9 and later versions (Guido van Rossum (2020)), the official parser transitioned away from LL(1). Note that such grammars can be approximated by removing certain features or imposing additional syntactic restrictions, though this often requires further workarounds and customized implementations.

Furthermore, although this issue is common across GCG methods, enforcing grammatical constraints often distorts the probability distribution produced by the LLM, making it difficult to sample text in a manner that faithfully reflects the model's original conditional probabilities under grammatical correctness. To address this, it is important to explore compatibility with methods that approximate the conditional distribution of LLMs under constraints, like Park et al. (2024).

# 7 CONCLUSION

In this paper, we proposed TruncProof, a novel LL(1)-constrained generation method designed to enable LLMs to produce grammatically valid outputs while adhering to a maximum token limit. Experiments on the Text-to-JSON instruction task (NousResearch, 2024) and Code generation task demonstrated that TruncProof can successfully generate syntactically correct outputs even under strict token constraints. We also show that TruncProof can be effectively combined with advanced decoding strategies, resulting in outputs that are not only grammatically valid but also semantically accurate. In future work, we plan to investigate methods to accelerate generation, particularly when using complex strategies. We also aim to extend our work to support general CFGs for broader applicability.

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

## A   THE USE OF LARGE LANGUAGE MODELS IN THIS PAPER

We used LLMs only to aid or polish writing.

## B   SUPPLEMENTARY MATERIAL

### B.1   DEFINITION OF LL(1) GRAMMAR

**Definition B.1** (LL(1) grammar). *A context-free grammar $(\mathcal{N}, \Sigma_T, R, S)$ is LL(1) grammar if, for all terminal sequences $w_1, w_2, w_2', w_3, w_3' \in \Sigma_T^\star$, a nonterminal $A \in \mathcal{N}$, and derivation rules $p, p' \in R$,*

$$\begin{cases} S \to^\star w_1 A w_3 \\ S \to^\star w_1 A w_3' \\ A \to^\star w_2 \text{ (The rule } p \text{ is applied first)} \\ A \to^\star w_2' \text{ (The rule } p' \text{ is applied first)} \\ (w_2.w_3) \text{ and } (w_2'.w_3') \text{ have the same prefix} \end{cases} \tag{7}$$

*implies $p = p'$.*

### B.2   SAMPLE PROMPT FOR JSON-MODE-EVAL

```
<bos><start_of_turn>user
You are a helpful assistant that answers in JSON. Here's the json schema you must adhere to:
<schema>
{'$id': 'https://example.com/entry-schema', '$schema': 'https://json-schema.org/draft/2020-12/
    schema', 'description': 'JSON Schema for an fstab entry', 'type': 'object', 'required':
    ['storage', 'fstype', 'options', 'readonly'], 'properties': {'storage': {'type': 'string
    ', 'pattern': '^/dev/[^/]+(/[^/]+)*$'}, 'fstype': {'type': 'string', 'enum': ['ext3', '
    ext4', 'btrfs']}, 'options': {'type': 'string', 'pattern': '^[a-zA-Z0-9,_-]+$'}, '
    readonly': {'type': 'boolean'}}}
</schema>
I need to define a JSON schema for a file system entry that includes specific constraints for
    the properties 'fstype', 'options', and 'readonly'. The 'fstype' should be limited to '
    ext3', 'ext4', or 'btrfs'. The 'options' should be a string that matches the pattern of
    comma-separated values, and 'readonly' should be a boolean indicating if the entry is
    read-only. Please provide me with a valid JSON object that adheres to these constraints.
    The file system entry should be for the storage '/dev/sda1', with 'fstype' as 'ext4', '
    options' set to 'rw,noatime', and 'readonly' as false.
Only output JSON.<end_of_turn>
```

### B.3   JSON GRAMMAR

```
?start: value

_BEGIN_ARR:   /[ \t\f\r\n]*\[[ \t\f\r\n]*/
_BEGIN_OBJ:   /[ \t\f\r\n]*\{[ \t\f\r\n]*/
_END_ARR:     /[ \t\f\r\n]*\][ \t\f\r\n]*/
_END_OBJ:     /[ \t\f\r\n]*\}[ \t\f\r\n]*/
_NAME_SEP:  /[ \t\f\r\n]*:[ \t\f\r\n]*/
_VALUE_SEP: /[ \t\f\r\n]*,[ \t\f\r\n]*/

?value: object
| array
| STRING
| number
| "true"            -> true
| "false"           -> false
| "null"            -> null

object: _BEGIN_OB [member (_VALUE_SEP member)*] _END_OBJ
member: STRING _NAME_SEP value
array : _BEGIN_ARR [value (_VALUE_SEP value)*] _END_ARR

number: MINUS? INT FRAC? EXP?
MINUS: "-"
INT: "0" | ("1".."9") DIGIT*
DIGIT: "0".."9"
FRAC: "." DIGIT+
```

```
EXP: ("e"|"E") ["+"|"-"] DIGIT+

STRING: /"([^"\\\x00-\x19]|\\["\\\/bfnrt]|\\u[0-9A-Fa-f]{4})*"/
```

## B.4    AN EXAMPLE OF CONTEXT-FREE GRAMMAR

For example, we consider the following CFG representing nested numbers list:

$$
\begin{aligned}
&\mathcal{N} = \{\langle\text{Expr}\rangle, \langle\text{Val}\rangle, \langle\text{Tail}\rangle\}, \ \Sigma = \{\text{Num}, [\,,\,]\,,\,;\,\} \\
&R = \left\{
\begin{array}{llll}
\langle\text{Expr}\rangle \to & [\ \langle\text{Expr}\rangle\ \langle\text{Tail}\rangle\ ] \\
\langle\text{Expr}\rangle \to & [\ \langle\text{Expr}\rangle\ ], & \langle\text{Expr}\rangle \to & \text{Num} \\
\langle\text{Tail}\rangle \to & ;\ \langle\text{Expr}\rangle\langle\text{Tail}\rangle, & \langle\text{Tail}\rangle \to & ;\ \langle\text{Expr}\rangle
\end{array}
\right. \\
&S = \langle\text{Expr}\rangle
\end{aligned}
\tag{8}
$$

Note that this definition is equivalent to the following Backus-Naur Form (BNF):

```
<Expr> ::= "[" <Expr> <Tail> "]"
         | "[" <Expr> "]"
         | <Num>
<Tail> ::= ";" <Expr> <Tail>
         | ";" <Expr>
```

For example, this CFG accepts a terminal sequence `[Num; [Num]]` because there is a derivation process described below.

$$
\begin{aligned}
\langle\text{Expr}\rangle &\to [\langle\text{Expr}\rangle\langle\text{Tail}\rangle] \to [\text{Num}\langle\text{Tail}\rangle] \to [\text{Num};\langle\text{Expr}\rangle] \\
&\to [\text{Num}; [\langle\text{Expr}\rangle]] \to [\text{Num}; [\text{Num}]]
\end{aligned}
\tag{9}
$$

We can visualize this derivation process as a derivation tree in Figure 6.

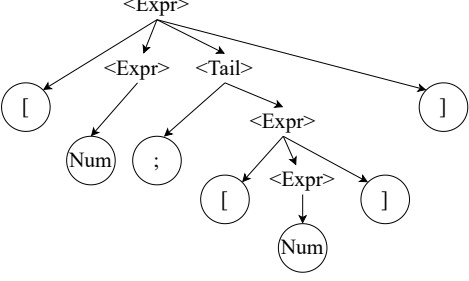

Figure 6: The derivation tree that represents the process in Equation 9.

## B.5 Our Algorithms in Detail

---

**Algorithm 1** Estimate shortest token length acceptable by a terminal's DFA

---

**Inputs:** $(Q_a, \Sigma, \delta_a, q_{a0}, F_a)$: DFA that accepts a terminal $a$, $Q_a^{live}$: a set of live states, $\mathcal{V}$: vocabulary
**Output:** the terminal's *lexical acceptance cost* $C_a[q] \in \mathbb{Z}_{\geq 0} \cup \{\infty\}$

1: Fill $C_a[q]$ with $\infty$ for all $q \in Q_a$
2: **for** each $q' \in Q_a^{live}$ **do**
3:     Fill $D[q]$ with $\infty$ for all $q \in Q_a$
4:     $D[q'] \leftarrow 0$
5:     $Q^{search} \leftarrow Q_a^{live}$
6:     **while** $Q^{search} \neq \emptyset$ **do**
7:         $u \leftarrow \arg\min_{u \in Q^{search}} D[u]$
8:         $Q^{search} \leftarrow Q^{search} - \{u\}$
9:         **for** each $t \in \mathcal{V}$ **do**
10:           $v \leftarrow \delta_a^*(u, t)$
11:           $D[v] \leftarrow \min(D[v], D[u] + 1)$
12:         **end for**
13:     **end while**
14:     $C_a[q'] \leftarrow \min_{q \in F_a} D[q]$
15: **end for**

---

**Algorithm 2** Approximate shortest token length derivable from a nonterminal

---

**Inputs:** $(\mathcal{N}, \Sigma_T, R, S)$: LL(1) grammar, $A$: nonterminal,
$C_a$: acceptance cost provided by Algorithm 1 for each $a \in \Sigma_T$
**Output:** the length of approximately shortest token sequence derivable from $A$
**Notation:** $A, B \in \mathcal{N}$, $\sigma, \tau \in \Sigma^*$, $\alpha, \alpha^{new}, \beta, \gamma_i, \delta \in (\mathcal{N} \cup \Sigma_T)^*$

1: Initialize $D$ as a map with default value $\infty$
2: $Q^{search} \leftarrow \{A\}$
3: $D[A] \leftarrow 0$
4: **while** true **do**
5:     $\alpha \leftarrow \arg\min_{\alpha \in Q^{search}} D[\alpha]$
6:     $Q^{search} \leftarrow Q^{search} - \{\alpha\}$
7:     **if** $\alpha$ is empty or all symbols in $\alpha$ are terminals **then**
8:         **return** $D[\alpha]$
9:     **end if**
10:     // Expand the leftmost nonterminal
11:     $\sigma, B\beta \leftarrow$ Split $\alpha$ into the leading terminals and the others
12:     **for** each rule $B \rightarrow \gamma_i$ in $R$ **do**
13:         $\alpha^{new} \leftarrow \sigma\gamma_i\beta$
14:         // Add costs of newly introduced leading terminals
15:         $\tau, \delta \leftarrow$ Split $\gamma_i\beta$ into the leading terminals and the others
16:         $d^{new} \leftarrow D[\alpha]$
17:         **for** each terminal $a$ in $\tau$ **do**
18:           $d^{new} \leftarrow d^{new} + C_a[q_{a0}]$
19:         **end for**
20:         $D[\alpha^{new}] \leftarrow d^{new}$
21:         $Q^{search} \leftarrow Q^{search} \cup \{\alpha^{new}\}$
22:     **end for**
23: **end while**

---

## B.6 Halting Problem of Algorithm 2

**Lemma B.1.** *Algorithm 2 always halts when the given grammar is LL(1).*

*Proof.* Let $G = (\mathcal{N}, \Sigma_T, R, S)$ be the given LL(1) grammar and $A$ be a nonterminal in $\mathcal{N}$. Assume there is a sentence $w \in \Sigma_T^*$ such that $A \to^* w$, and there is no terminal which allows an empty string, i.e. $C_a[q_{a0}] > 0$ for all $a \in \Sigma_T$. With this assumption, when the number of leading terminals in a sequence $\alpha^{new}$ increases, the cost $D[\alpha^{new}]$ increases monotonically. On the other hand, in some finite derivation steps, the number of leading terminals increases monotonically because LL(1) grammars don't accept the left-recursion $B \to^* B\beta$ (Lemma 8.3 in Aho & Ullman (1972)) and a set of nonterminals is finite. Therefore, for any cost $d$, the number of the possible derivation $\alpha$ from $A$ with $D[\alpha] < d$ is finite. This means the algorithm finds $w$ with $D[w]$ and halts in some finite iterations of the while-loop. $\qquad \square$

### B.7 GUARANTEE OF TRUNCPROOF

**Lemma B.2.** *Our constraint mask guarantees grammatically correct output shorter than the specified limit $N_{max}$.*

*Proof.* Assume that we have selected the token $t_i$ based on the constraint mask in iteration $i$, and the intermediate output becomes $t_{<i}.t_i$. At that time $t_{<i}$ is divided into the terminal sequence $\tau \in \Sigma_T^*$ and the reminder $r$, and there is an accept sequence $(a, b)$ that holds:

$$i + C_{a+b}[\delta_{a+b}^*(q_{a+b0}, r.t_i)] + d_{cost}(\tau.a.b) < N_{max} \tag{10}$$

and there are three possibilities.

**(A)** When $C_{a+b}[\delta_{a+b}^*(q_{a+b0}, r.t_i)] = d_{cost}(\tau.a.b) = 0$, the intermediate output completes the grammatically correct string, so we can stop generation or optionally output EOS. The generated result is grammatically correct and meets the token limit because $i < N_{max}$.

**(B)** When $C_{a+b}[\delta_{a+b}^*(q_{a+b0}, r.t_i)] > 0$, there is a token $t$ that holds:

$$C_{a+b}[\delta_{a+b}^*(q_{a+b0}, r.t_i.t)] \leq C_{a+b}[\delta_{a+b}^*(q_{a+b0}, r.t_i)] - 1 \tag{11}$$

Based on Equation 10,

$$i + 1 + C_{a+b}[\delta_{a+b}^*(q_{a+b0}, r.t_i.t)] + d_{cost}(\tau.a.b) < N_{max} \tag{12}$$

This means $m_t^{(a,b)} = true$ in iteration $i + 1$.

**(C)** When $C_{a+b}[\delta_{a+b}^*(q_{a+b0}, r.t_i)] = 0$ and $d_{cost}(\tau.a.b) > 0$, the intermediate output $t_{<i}.t_i$ is divided into $\tau.a.b$ and there is a sequence of terminals $\sigma_1. \cdots \sigma_k$ where $\tau.a.b.\sigma_1. \cdots \sigma_k \in L(G)$ and $\sum_{j=1}^k C_{\sigma_j}[q_{\sigma_j 0}] = d_{cost}(\tau.a.b)$. Note that $k \geq 1$ because $C_{\sigma_j}[q_{\sigma_j 0}] > 0$ for all $j$. Therefore, it holds:

$$i + C_{\sigma_1}[q_{\sigma_1 0}] + d_{cost}(\tau.a.b.\sigma_1) < N_{max} \tag{13}$$

Because $C_{\sigma_1}[q_{\sigma_1 0}] > 0$, there is a token $t$ that holds:

$$i + 1 + C_{\sigma_1}[\delta_{\sigma_1}^*(q_{\sigma_1 0}, t)] + d_{cost}(\tau.a.b.\sigma_1) < N_{max} \tag{14}$$

This means $m_t^{(\sigma_1)} = true$ in iteration $i + 1$.

Therefore, we can continue to build valid constraint masks throughout text generation and can stop the generation once condition (A) holds. $\qquad \square$

## B.8 Experiments on JSON-Mode-Eval under the token limit provided by Ugare et al. (2024)

Table 2: Accuracy of JSON-Mode-Eval under the original token limit 400.

| Model | Method | Decoding | Accuracy (%) Syntax | Schema | Exact-match |
|---|---|---|---|---|---|
| | No constraint | Greedy | 38 | 38 | 29 |
| | Outlines | Greedy | **100** | 96 | 72 |
| | SynCode | Greedy | 99 | 97 | 73 |
| | XGrammar | Greedy | 99 | **99** | **74** |
| Gemma2-2B | Ours | Greedy | **100** | 95 | 72 |
| | No constraint | Greedy | 6 | 5 | 0 |
| | Outlines | Greedy | **100** | **67** | **45** |
| | SynCode | Greedy | 98 | 61 | 40 |
| | XGrammar | Greedy | 98 | 44 | 26 |
| Llama2-7B-Chat-HF | Ours | Greedy | **100** | 63 | 40 |

## B.9 C Grammar specified in Figure 5

```
start: declaration*

declaration: data_type NAME "(" parameters? ")" "{" statement* "}"

statement: data_type NAME "=" expression ";"
 | NAME  "="  expression ";"
 | NAME  "(" arg_list? ")" ";"
 | "return"  expression ";"
 | "while" "(" condition ")" "{" statement* "}"
 | "for" "(" for_init ";"  condition ";"  for_update ")" "{" statement* "}"
 | "if" "(" condition ")" "{" statement* "}" ("else" "{" statement* "}")?

data_type: "int" | "float"  | "char" | "void"
NAME: /[a-zA-Z_][a-zA-Z_0-9]*/

parameters: parameter ("," parameter)*
parameter: data_type NAME

for_init: data_type NAME  "="  expression | NAME  "="  expression
for_update: NAME  "="  expression

condition: expression relation_operator expression
relation_operator: ("<=" | "<" | "==" | "!=" | ">=" | ">")

expression: term (("+" | "-") term)*
term: factor(("*" | "/") factor)*

factor: NAME | number | unary_term | NAME "(" arg_list? ")" | paren_expr
unary_term: "-" factor
paren_expr: "("  expression  ")"

arg_list: expression (","  expression)*

number: /[0-9]+/

WS : /[ \t\n]+/
%ignore WS
```

## B.10 Ranging expansion ratios

Figure 7 presents the results with different expansion ratios, *i.e.*, $e \in [1.0, 1.5]$. We observe that our method consistently adheres to the instructed schema, even under strict maximum token limits. Moreover, when combined with BS or MCTS, our approach preserves the correctness of the generated content across various expansion settings. These results experimentally validate the ef-

fectiveness of TruncProof in generating grammatically correct outputs, as well as its compatibility with various decoding strategies, which leads to improved semantic quality of the generated texts.

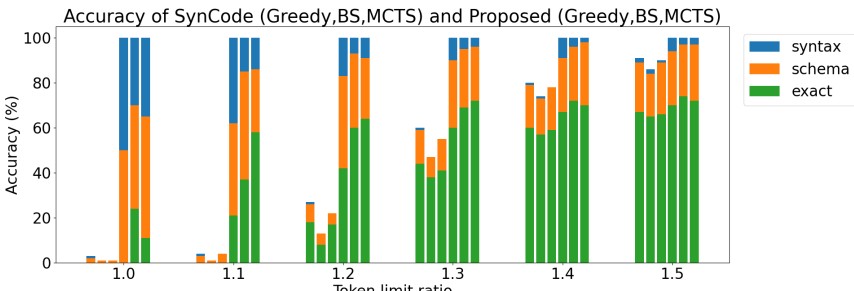

Figure 7: Accuracy of Gemma2-2B with respect to the expansion ratio $e \in [1.0, 1.5]$. Six bars drawn in each ratio are the results of SynCode with Greedy decoding, SynCode with Beam Search, SynCode with Monte Carlo Tree Search, ours with Greedy decoding, ours with Beam Search and ours with Monte Carlo Tree Search.

### B.11 Accuracy of JSON-mode-eval with prompt engineering

To compare the shortening effect of prompt engineering with TruncProof's capabilities, we add the prompt *"Only output JSON. Eliminate white spaces and keep the output as compact as possible."* to the original prompt provided by JSON-Mode-Eval. Results are shown as +*prompt* in Table 3 and Table 4. This additional prompt improves the performance slightly in several settings. As a side effect, unnecessary text such as ```` ```json ```` is less frequent, leading to a certain degree of gains in the absence of grammar constraints ("No Constraint" rows). However, it was challenging to ensure LLMs adhere to the maximum token limit when relying solely on prompts.

Table 3: Accuracy of JSON-Mode-Eval under the token limit 400.

| Model | Method | Decoding | Accuracy (%) Syntax | Schema | Exact-match |
|---|---|---|---|---|---|
| | No constraint | Greedy | 38 | 38 | 29 |
| | No constraint +*prompt* | Greedy | 79 | 78 | 59 |
| | SynCode | Greedy | 99 | 97 | **73** |
| | SynCode +*prompt* | Greedy | **100** | 98 | 72 |
| | Ours | Greedy | **100** | 95 | 72 |
| Gemma2-2B | Ours +*prompt* | Greedy | **100** | **99** | 72 |
| | No constraint | Greedy | 6 | 5 | 0 |
| | No constraint +*prompt* | Greedy | 6 | 6 | 2 |
| | SynCode | Greedy | 98 | 61 | 40 |
| | SynCode +*prompt* | Greedy | 95 | 73 | **49** |
| | Ours | Greedy | **100** | 63 | 40 |
| Llama2-7B-Chat-HF | Ours +*prompt* | Greedy | **100** | **76** | 48 |

Table 4: Accuracy of JSON-mode-eval with $e = 1.1$.

| Model | Method | Decoding | Accuracy (%) | | |
|---|---|---|---|---|---|
| | | | Syntax | Schema | Exact-match |
| | No constraint | Greedy | 1 | 1 | 0 |
| | No constraint +*prompt* | Greedy | 8 | 8 | 4 |
| | SynCode | Greedy | 4 | 3 | 0 |
| | SynCode +*prompt* | Greedy | 6 | 6 | 1 |
| | SynCode | BS | 1 | 1 | 0 |
| | SynCode +*prompt* | BS | 2 | 2 | 0 |
| | Ours | Greedy | **100** | 62 | 21 |
| | Ours +*prompt* | Greedy | **100** | 68 | 12 |
| | Ours | BS | **100** | 85 | 37 |
| | Ours +*prompt* | BS | **100** | 84 | 45 |
| | Ours | MCTS | **100** | 86 | 58 |
| Gemma2-2B | Ours +*prompt* | MCTS | **100** | **90** | **65** |
| | No constraint | Greedy | 2 | 2 | 0 |
| | No constraint +*prompt* | Greedy | 2 | 2 | 0 |
| | SynCode | Greedy | 11 | 10 | 4 |
| | SynCode +*prompt* | Greedy | 16 | 14 | 5 |
| | SynCode | BS | 6 | 6 | 4 |
| | SynCode +*prompt* | BS | 13 | 12 | 5 |
| | Ours | Greedy | **100** | 51 | 2 |
| | Ours +*prompt* | Greedy | **100** | 57 | 2 |
| | Ours | BS | **100** | 67 | 29 |
| | Ours +*prompt* | BS | **100** | 68 | 32 |
| Llama2-7B | Ours | MCTS | **100** | **70** | **41** |
| -Chat-HF | Ours +*prompt* | MCTS | **100** | **70** | **41** |

