# OpenReview forum: "TruncProof: LL(1)-Constrained Generation in Large Language Models with Maximum Token Limitations"
_ICLR.cc/2026/Conference — Submitted to ICLR 2026_

### Official Review · Reviewer_AGnR · 2025-10-29

**Soundness:** 3
**Presentation:** 2
**Contribution:** 3
**Rating:** 4
**Confidence:** 4

**Summary:**

This paper augments constrained-decoding for LL(1) grammars to take into account a length constraint on the numbers of tokens. Given a prefix and a next token, one wants to only continue with that token if there exists a valid grammatical completion that uses only as many tokens as the length constraint allows. In general this problem is decidable but one has to make some tradeoffs with efficiency to avoid expensive computations that will cause masking to be expensive. The key observation of the paper is that for LL(1) computing a bound on how many tokens are needed to complete a sequence is pretty easy as the stack of an LL(1) parser tells us exactly how to do that (we just need to take the sum of all the shortest completions of each nonterminal/terminal on the stack). The approach is evaluated on very restrictive grammars to illustrate how normal constrained decoding would (unsurprisingly) fail to abide to the limited number of allowed tokens.

**Strengths:**

- I think the problem of dealing with a bounded number of tokens is a very good problem to study as this is the main failure mode of constrained decoding approaches; they tend to go on and on generating non-sense just to follow the constraints.
- I like the observation that for LL(1) whether one can stay within the bound is easier to compute
- The results are encouraging at showing that this approach provides some flexibility in setting bounds on numbers of tokens

**Weaknesses:**

- LL(1) is a very limited set of grammars. Most of the grammars used in existing GCD papers (e.g., https://icml.cc/virtual/2025/poster/45613) are not LL(1).

- It is a bit disappointing that the paper, despite targeting a small fragment (LL(1)) still computes an overapproximation of how many tokens a completion will result in (specifically the work doesn't consider that LLM tokens may span across different PL tokens (e.g., if an LLM uses the token  *b")*, which spans a literal, a quote, and a parenthesis, eq(5) will count this token at least twice, once for completing the literal, and once for the tree completion

- Setting the token budget to 1.1X the ground truth is a bit of an arbitrary evaluation.

- The paper doesn't report the preprocessing cost of their tool (other papers who have done so in the past, had very high preprocessing costs)

- The tokens/sec reported in Table 1 are not for Sota tool. Should consider, for example, llguidance


Other comments:

The paper is missing many of the more current works for constrained decoding. For SOTA methods, they should really cite LLGuidance (https://github.com/guidance-ai/llguidance) since it squarely defeats XGrammar and other tools across all metrics, and GreatGramma (https://icml.cc/virtual/2025/poster/45613) since it supports the most complex grammars while guaranteeing reasonable processing times.

I don't understand the point at line 121-122 distinguishing top down and bottom up parsers. I feel like the problem won't be much harder for LR(1) parser. One can use some dynamic programming to compute the shortest completion possible.

**Questions:**

- What is the preprocessing cost of the tool?
- Line 374 mentions "As discussed in... TruncProof with grreedy decdoing does not fully account for semantic correctness". What does this mean? This aspect is not discussed
- Table 1 mentions 100 syntax correct for Greedy/Ours, but not 100 for schema/exact. Shouldn't Greedy generate always the same output?
- The fact that TruncProof proposes the Exact-match so many times is a bit suspicious and makes one think the search space is designed for the tool to do so. What happens with e=2?

---

> ### Author Response · Authors · 2025-11-20
> **Response to the official review**
>
> Thank you for reviewing our paper. The response became lengthy, so we divided it into two posts.
>
> ## Extension for LR(1)
>
> > LL(1) is a very limited set of grammars.
>
> We agree that LL(1) is weak to practical programming languages, but we think that LL(1) has sufficient flexibility for a sort of user-defined grammars that defines some data structure, such as JSON and TOML.
>
> > I feel like the problem won't be much harder for LR(1) parser. One can use some dynamic programming to compute the shortest completion possible.
>
> Thank you for your insightful suggestion.
>
> However, we think there is unlikely to be any more efficient algorithm (such as DP) than brute-forcing terminals (Line 281) for LR(1) because of the following two reasons.
>
> First, in generation, the optimal next derivation depends on the already derived subtrees (e.g., node ‘a’ and a subtree with root ‘S’ in Figure 1(a)), and they cannot be resolved independently (e.g., shortest continuation from S is not always the prefix of the shortest continuation from aS).
>
> Second, in preprocess, it is infeasible to estimate the shortest terminal sequence derivable from a nonterminal, because there are infinite number of derivation processes and CFG allows recursive derivation without introducing any terminal.
>
> ## overapproximation of token length
>
> > the work doesn't consider that LLM tokens may span across different PL tokens…which spans a literal, a quote, and a parenthesis, eq(5) will count this token at least twice, once for completing the literal, and once for the tree completion
>
> As you may recall, our method counts an LLM token that spans across multiple dangling symbols (Figure 2(c)) at least twice, which leads to the “overapproximation” you stated.
> TruncProof masks out some tokens despite they are valid because of these reasons:
>
> - As you stated, the number of future tokens $d_{cost}$ in eq(5) tends to be larger than the optimal continuation.
> - TruncProof rejects the tokens which spans three terminals (but they are rare in the vocabulary).
>
> However, we want to emphasize that there is no underapproximation which allows dangerous tokens, and the critical degradation of LLM’s fluency is avoided by dealing with two-terminal tokens in masking (Line 215 and footnote2).
>
> ## more previous works
>
> > The tokens/sec reported in Table 1 are not for Sota tool. Should consider, for example, llguidance
>
> > The paper is missing many of the more current works for constrained decoding.
>
> As far as we understand, both llguidance and GreatGramma primarily aim to reduce computational cost in preprocessing or runtime, and their behavior under token limitations is expected to be similar to that of other existing methods. Therefore, we believe that our experiments (Line 372-420, Table 1 and Figure 3(b)) sufficiently demonstrate the superiority of our method: the robustness under strict token limitations.
>
> Although conducting experiments on these datasets would be valuable, we unfortunately leave them for future work due to space limitations. We will cite llguidance & GreatGramma in the Related Works section in the next revision. Thank you for your suggestion.
>
> ## preprocessing cost
>
> > What is the preprocessing cost of the tool?
>
> In the current manuscript (Line 353), we report the preprocessing time for JSON (1 min) and C (5 min). The preprocessing time tends to be proportional to $|\Sigma_T|^2$, which is the number of DFAs we have to prepare (related to Eqn 3 and precomputing the mapping of DFAs). It is also affected by the number of nonterminal symbols in the grammar (related to Eqn 4).
>
> ## “As discussed in... TruncProof with greedy decoding does not fully account for semantic correctness”
>
> We didn't explicitly discuss this in the paper. Thank you for pointing it out.
>
> Our method only considers the grammar and the number of future tokens, so it allows grammatically correct but semantically unnatural outputs.
> For example, our TruncProof prohibits (2) but allows (3) in the following JSON outputs.
>
> ```
> (1) (ground truth) { "ssid": "OfficeNet", "protocol": "WPA2-Enterprise", "bandwidth": "1300 Mbps" }
> (2) (truncated and grammatically incorrect) { "ssid": "OfficeNet", "protocol": "WPA2-Enterprise", "bandwidth": "1
> (3) (grammar is OK but incorrect) { "ssid": "OfficeNet", "protocol": "WPA2-Enterprise", "bandwidth": "" }
> ```
>
> But when the essential parts are already generated within the budget, TruncProof helps the accuracy of content (it is rare case though).
>
> ```
> SynCode Greedy (grammatically incorrect):
> {
> "data": [1, 2, 3, 4, 5]
>
> TruncProof Greedy (exact-match):
> {
> "data": [1, 2, 3, 4,5] }
> ```
>
> Another example is shown in Figure 4. The C code generated by TruncProof (PPL 70.0) is compilable, but this function is meaningless.
> To improve the semantics, we combine TruncProof with BeamSearch or MCTS and we show it works.

---

> > ### Author Response · Authors · 2025-11-20
> > **(Continued) Response to the official review**
> >
> > ## 100 Syntax correct for Greedy but not for Schema/Exact-Match
> >
> > The reason is same as the above section. Correct syntax is not equivalent to correct schema. Here is the example of exact-match and incorrect outputs.
> >
> > ```
> > Exact-match:
> > { "ssid": "OfficeNet", "protocol": "WPA2-Enterprise", "bandwidth": "1300 Mbps" }
> >
> > Schema (but not Exact-match):
> > { "ssid": "OfficeNet", "protocol": "WPA2-Enterprise", "bandwidth": "" }
> >
> > Syntax (but not Schema)
> > { "ssid": "OfficeNet", "protocol": "WPA2-Enterprise" }
> > (“bandwidth”: str entry is missing)
> > ```
> >
> > ## What happens with e=2
> >
> > > Setting the token budget to 1.1X the ground truth is a bit of an arbitrary evaluation.
> >
> > We set the token budget to e=1.1 times the ground truth because we wanted to design the evaluation to reach the token limit when the LLM outputs unnecessary whitespace (LLMs usually do it for readability).
> >
> > For easier limitations, we additionally conducted the experiment that uses the setup of $e\in\{1.0, 1.1, 1.2, 1.3, 1.4, 1.5\}$ in Appendix B10.  We observed the previous method SynCode drops accuracy as the token limit gets shorter even with Beam Search or MCTS.
> >
> > **NOTE:** Our experiment script for Figure 6 in Appendix B10 had a bug and the values with MCTS  are incorrect. Here is the corrected metrics:
> >
> > | Method     | Strategy   |   e=1.00 |   e=1.10 |   e=1.20 |   e=1.30 |   e=1.40 |   e=1.50 |
> > |:-----------|:-----------|---------:|---------:|---------:|---------:|---------:|---------:|
> > | SynCode    | Greedy     |        0 |        0 |       18 |       44 |       60 |       67 |
> > | SynCode    | BS         |        0 |        0 |        8 |       38 |       57 |       65 |
> > | SynCode    | MCTS       |        0 |        0 |       17 |       41 |       59 |       66 |
> > | TruncProof | Greedy     |        0 |       21 |       42 |       60 |       67 |       70 |
> > | TruncProof | BS         |       24 |       37 |       60 |       69 |       72 |       74 |
> > | TruncProof | MCTS       |       11 |       58 |       64 |       72 |       70 |       72 |

---

> > ### Comment · Reviewer_AGnR · 2025-11-26
> >
> > > Although conducting experiments on these datasets would be valuable, we unfortunately leave them for future work due to space limitations
> >
> > The grammars used in those benchmarks are not LL(1) so I wouldn't say this is an issue of space limitations. The paper should clearly describe what grammars used in prior work are not LL(1) to warn the user about the applications one can't reach.
> >
> > > First, in generation, the optimal next derivation depends on the already derived subtrees (e.g., node ‘a’ and a subtree with root ‘S’ in Figure 1(a)), and they cannot be resolved independently (e.g., shortest continuation from S is not always the prefix of the shortest continuation from aS).
> >
> > Yes, but one can use any grammar emptiness algorithm to find the shortest such sequence. The prefix is a regular language, you can compute the intersection of the grammar with the prefix in poly time and then run grammar emptiness to find the shortest completion. If one is willing to accept O(n^3) in the length of the string, this can be done for any grammar, not even just LR.
> >
> > > Second, in preprocess, it is infeasible to estimate the shortest terminal sequence derivable from a nonterminal, because there are infinite number of derivation processes and CFG allows recursive derivation without introducing any terminal.
> >
> > Why is this true? This is how any grammar emptiness algorithm works, no algorithm explores "infinitely many derivations". This feels like saying that finding the shortest path in a graph is infeasible because the graph may have loops.
> > For a grammar S -> ( S ) | (), we know the shortest sequence is () and this can be computed via a simple least fixed point algorithm.

---

> > > ### Author Response · Authors · 2025-11-27
> > >
> > > > > Although conducting experiments on these datasets would be valuable, we unfortunately leave them for future work due to space limitations
> > > >
> > > > The grammars used in those benchmarks are not LL(1) so I wouldn't say this is an issue of space limitations. The paper should clearly describe what grammars used in prior work are not LL(1) to warn the user about the applications one can't reach.
> > >
> > > For the experiments on programming languages and SQL in previous works, we will explicitly note that TruncProof does not support them due to expressiveness limitations in LL(1) grammars in the revised manuscript.
> > >
> > >
> > > > > Second, in preprocess, it is infeasible to estimate the shortest terminal sequence derivable from a nonterminal, because there are infinite number of derivation processes and CFG allows recursive derivation without introducing any terminal.
> > > >
> > > > Why is this true? This is how any grammar emptiness algorithm works, no algorithm explores "infinitely many derivations". This feels like saying that finding the shortest path in a graph is infeasible because the graph may have loops. For a grammar S -> ( S ) | (), we know the shortest sequence is () and this can be computed via a simple least fixed point algorithm.
> > >
> > > It makes sense. We can use depth-first search to find the shortest sequence for each nonterminal with a visited set of nonterminals.
> > >
> > > > > First, in generation, the optimal next derivation depends on the already derived subtrees (e.g., node ‘a’ and a subtree with root ‘S’ in Figure 1(a)), and they cannot be resolved independently (e.g., shortest continuation from S is not always the prefix of the shortest continuation from aS).
> > > >
> > > > Yes, but one can use any grammar emptiness algorithm to find the shortest such sequence. The prefix is a regular language, you can compute the intersection of the grammar with the prefix in poly time and then run grammar emptiness to find the shortest completion. If one is willing to accept O(n^3) in the length of the string, this can be done for any grammar, not even just LR.
> > >
> > > We are still unsure that the time complexity is O(N^3), but we understand that it will likely in polynomial.
> > > Does it mean CFG spends more time searching shortest sequences than TruncProof does in runtime? LL(1) requires $O(|\Gamma|)$ to estimate the minimum number of tokens thanks to the precomputation, but it seems CFG needs much more computation because an unseen start symbol is introduced in calculating the intersection of the CFG and a prefix.

---

> > > > ### Comment · Reviewer_AGnR · 2025-11-27
> > > >
> > > > > Does it mean CFG spends more time searching shortest sequences than TruncProof does in runtime?
> > > >
> > > > Yes, general CFGs require non-linear algorithms. But this is already the case if one even wants do regular constrained decoding for an arbitrary CFG. There are no linear-time parsing algorithms. This is why people invented LL/LR/LALR in the first place

---

### Official Review · Reviewer_wgCt · 2025-10-29

**Soundness:** 2
**Presentation:** 3
**Contribution:** 2
**Rating:** 4
**Confidence:** 3

**Summary:**

This paper addresses a problem in grammar-constrained generation with token limits. When prior grammar-constrained generation techniques, such as syncode and xgrammar, reach the maximum token count, they stop generating, which often results in incomplete or invalid outputs. The paper proposes TruncProof, which uses LL(1) parsers to calculate at each generation step how many tokens are minimally required to complete a grammatically valid output. This allows the method to block token selections that would either break the grammar or exceed the token budget. The approach is implemented as a logit modifier, making it compatible with different models and decoding methods. Experiments on JSON generation and C code tasks demonstrate that TruncProof maintains grammatical validity under strict token constraints, while baseline methods largely fail in these scenarios.

**Strengths:**

* The paper is well-written and easy to read
* I appreciate the time complexity and space complexity analysis. It would be better if the authors could compare these complexities with prior works.
* The related work section is comprehensive and considers most SOTA CFG constrained generation works.
* The evaluation considers various sampling techniques (beam search, MCTS) during inference. In my knowledge, this has not been studied extensively in the prior CFG-constrained decoding works.

**Weaknesses:**

The major issue in the paper is the limited empirical evidence showing the practical applicability of the work beyond JSON generation under token-limit restriction. I would encourage the authors to perform additional experiments on one of the other grammars such SQL, Python, Java, or Go, that have been considered in prior works and use end-to-end benchmarks such as humaneval.

* The technique uses LL(1) grammar, which is inherently weaker than LR and Earley grammars supported by some of the prior works  .

* The evaluation on JSON considers only 100 examples on 2 models. Both should ideally be more.

* The evaluation for C code generation is almost non-existent. Is it just evaluated on a single example in Figure 4? Am I missing something?

**Questions:**

* What is the motivation for the token limit restriction?

* In case of JSON generation, LLM generates some extra whitespaces/newlines which are avoidable but is it also realistic in other programming languages?

---

> ### Author Response · Authors · 2025-11-20
> **Response to the official review**
>
> Thank you for reviewing our paper.
>
> ## Motivation of token limitation
>
> As we briefly mentioned in Line 041, LLMs sometimes generate endless output, and it is unforeseeable and practically unmanageable solely through prompts. LLMs don’t recognize the number of their output tokens, and prompts like “keep the output as compact as possible” aren’t strictly followed (See Appendix B.11 Table 4 for the experiments on such prompts).
>
> It is not problematic in the use-cases of chatbot because users rarely care about it even if we interrupt the generation. However, in scenarios of agent-based applications where structured output generated by an agent is directly fed to another agent, truncated output causes parse errors that subsequently break downstream processes.
>
> ## Support for more complex grammars
>
> > The technique uses LL(1) grammar, which is inherently weaker than LR and Earley grammars supported by some of the prior works.
>
> We agree that LL(1) is weak to practical programming languages, but we think that LL(1) has sufficient flexibility for a sort of user-defined grammars that defines some data structure, such as JSON and TOML.
>
> > The evaluation on JSON considers only 100 examples on 2 models. Both should ideally be more.
>
> > In case of JSON generation, LLM generates some extra whitespaces/newlines which are avoidable but is it also realistic in other programming languages?
>
> Existing methods (SynCode, XGrammar) used JSON-Mode-Eval, and in this benchmark we demonstrate the superiority of our TruncProof in our scope of research: a robustness under the limited number of tokens.
>
> We agree that the JSON generation is easy for recent LLMs and there should be some quantitive evaluations on more complex grammars. However, our research scope addresses the challenging problem that even simple constrained generation becomes unstable when token limits are imposed, and we think TruncProof successfully addresses this issue.
>
> > The evaluation for C code generation is almost non-existent. Is it just evaluated on a single example in Figure 4? Am I missing something?
>
> We include a single example in Figure 4 to mention the possibility that TruncProof with MCTS could achieve more than just removing whitespace to preserve semantics.
>
> > The evaluation … on 2 models.
>
> As we described in Line 057, TruncProof has a form of logit modifier (called ”LogitsProcessor by HuggingFace), which is supported by nearly all modern LLMs. So we think that evaluations on two LLMs chosen by SynCode are sufficient.

---

> > ### Comment · Reviewer_wgCt · 2025-11-26
> >
> > I have read the author's response. The response does not sufficiently address my concerns regarding the motivation and issues with the current evaluation. Hence, I will keep my current score.

---

### Official Review · Reviewer_64zc · 2025-10-31

**Soundness:** 4
**Presentation:** 3
**Contribution:** 3
**Rating:** 8
**Confidence:** 4

**Summary:**

The paper presents TruncProof. It allows for strict LLM generation adherence to the token-budget, while still generating syntactically correct outputs. The constrained generation approach is based on LL(1) parsing of context free grammars. The authors performed experiments on Text-to-JSON instructions and code generation tasks, improving over the existing approaches such as Outlines, Syncode, and XGrammar. The approach enhances the semantic robustness of the JSON and C output by leveraging decoding strategies such as Beam Search and Monte Carlo Tree Search.

**Strengths:**

* Interesting and potentially useful technique for controlling LLM output with strict token budget.
* Technical approach is intuitive and effective.
* The experimental results show that TruncProof is effective on challenging function calling/coding tasks under strict token budget.
* The paper is well written.

**Weaknesses:**

* The experiments are performed only for two budget threshold. A better understanding of the tradeoff between the number of tokens and the syntactic&semantic correctness would be welcome.
* The approach may over constrain the semantic generation.

**Questions:**

The paper presents an interesting practical technique for controlling the output of LLMs. While existing constrained decoding algorithms have shown how to ensure the LLMs only generate legal tokens, they don’t have means to ensure that the output will be properly generated in a fixed amount of time (tokens). Thus, many LLM generations in JSON/coding tasks have occurred because the model was not able to close all scopes etc.

The approach is intuitive, yet demonstrated to be effective. The algorithm computes the approximate shortest token length derivable from each nonterminal ahead of time. It further track the number of tokens consumed and can adjust the rest of generation to follow the path until guaranteed completion. The authors show that this approach  has reasonable worst-case time and space complexity, although I would have appreciated if it also included the estimates of expected time/space consumption (and relate the algorithm properties with the grammars they used in the evaluation).

The experiments compare TruncProof with several state-of-the-art tools for constrained decoding. The results show the significant improvement in syntactically correct codes under a tight token budget. However, the paper presents only two (the aggressive in the paper; the permissive in the appendix) limit. It is important to understand the tradeoffs for multiple token budget thresholds, between the two extremes presented in the paper and in the appendix. Showing these additional results with even one selected decoding strategy would be valuable.

Further it would be interesting to see the behavior of the approach on a reasoning LLMs and how token budgets impact their performance. Overall, with strict constraining it may be possible the model yields suboptimal results regarding semantic correctness.

Finally, I enjoyed reading the paper and found it well-written.

---

> ### Author Response · Authors · 2025-11-20
> **Response to the official review**
>
> Thank you for reviewing our paper.
>
> ## Expected time / space complexity
>
> > I would have appreciated if it also included the estimates of expected time/space consumption (and relate the algorithm properties with the grammars they used in the evaluation).
>
> First, the space complexity (Line 284-287) is fixed once the grammar is specified.
> Second, the time complexity (Line 270-283) changes because the accept sequence (defined at Line 238) changes during generation. So the time complexity can be described as $O(|\mathcal{A}|(T_G+|\Gamma|))$. We will note it in the revised version.
>
> We measured the parameters during the evaluation on the JSON-Mode-Eval dataset:
>
> - the number of terminals $|\Sigma_T| = 15$ (It is described in Line 275)
> - the number of DFAs: 73 (it is usually smaller than $|\Sigma_T|^2$ because some terminal sequences are impossible)
> - the size of DFA states $|Q|$:  min 2, average 4.97, max 9
> - the number of acceptance sequences $|\mathcal{A}|$: min 1, average 13.4, max 26 (upper bound is 73)
> - the number of dangling symbols $|\Gamma|$: min 1, average 3.46, max 9
>
> From these measurements, we think the size of $\mathcal{A}$ is the bottleneck of both preprocessing and runtime.
>
> ## Range of token limit
>
> > It is important to understand the tradeoffs for multiple token budget thresholds, between the two extremes presented in the paper and in the appendix.
>
> In Appendix B10 of our current manuscript, we present the results of SynCode and our TruncProof while gradually varying the limitation. We observed the previous method SynCode drops accuracy as the token limit gets shorter, but the degradation of TruncProof is mitigated.
>
> **NOTE:** Our experiment script for Figure 6 had a bug and the values with MCTS are incorrect. Here is the corrected metrics of Exact-Match with respect to expansion ratios:
>
> | Method     | Strategy   |   e=1.00 |   e=1.10 |   e=1.20 |   e=1.30 |   e=1.40 |   e=1.50 |
> |:-----------|:-----------|---------:|---------:|---------:|---------:|---------:|---------:|
> | SynCode    | Greedy     |        0 |        0 |       18 |       44 |       60 |       67 |
> | SynCode    | BS         |        0 |        0 |        8 |       38 |       57 |       65 |
> | SynCode    | MCTS       |        0 |        0 |       17 |       41 |       59 |       66 |
> | TruncProof | Greedy     |        0 |       21 |       42 |       60 |       67 |       70 |
> | TruncProof | BS         |       24 |       37 |       60 |       69 |       72 |       74 |
> | TruncProof | MCTS       |       11 |       58 |       64 |       72 |       70 |       72 |
>
> ## Degradation of semantic correctness under strict constraints
>
> > Further it would be interesting to see the behavior of the approach on a reasoning LLMs and how token budgets impact their performance. Overall, with strict constraining it may be possible the model yields suboptimal results regarding semantic correctness.
>
> Not limited to token budget, constrained generation distorts the probability provided by LLMs, which harms their fluency or changes their behavior of sampling (as described in Line 470).

---

> > ### Comment · Reviewer_64zc · 2025-11-28
> >
> > Thank you for the additional results. I remain positive about the paper and think it is useful technique.

---

### Official Review · Reviewer_V4hk · 2025-11-01

**Soundness:** 3
**Presentation:** 2
**Contribution:** 2
**Rating:** 4
**Confidence:** 4

**Summary:**

This paper introduces TruncProof, a grammar-constrained decoding method that leverages LL(1) parsers to estimate the minimum tokens needed to complete a valid output at each step, enabling LLMs to generate syntactically valid outputs within a fixed token limit.

**Strengths:**

The RQ is clear, and the paper is relatively easy to follow.

**Weaknesses:**

Conceptual clarity and motivation: The paper would benefit from clearer motivation and explanation of why enforcing token limits is necessary and how it improves decoding quality, e.g., does it improve overall accuracy or just ensure early termination? It is not fully clear why adding an LL(1)-based token constraint offers an advantage over simply applying standard grammar-constrained decoding (e.g., CFG-based methods) with an explicit token or context budget specified in the prompt. The practical benefit, especially when existing constrained decoding already guarantees syntactic validity, remains ambiguous.

Writing: Providing a concrete, step-by-step example walkthrough of each phase of TruncProof (rather than only one abstract output example in Fig. 4) would greatly improve clarity and help readers understand how the approach operates and why it matters.

Experimental limitations:
The choice of baselines is limited and may not fully isolate TruncProof’s contribution. A stronger comparison would include variants that apply existing grammar constraints or prompt-level token limits under the same decoding setup.
Additionally, the overhead introduced by TruncProof, both in preprocessing and runtime, especially whether scaling up grammar size will affect decoding efficiency, should be quantified and discussed.
Finally, the evaluation focuses primarily on syntactic validity rather than overall task performance; adding metrics for semantic correctness or downstream accuracy (e.g., in code completion, data-to-JSON generation, or semantic parsing) would help clarify whether the proposed constraint mechanism provides real benefits beyond ensuring grammaticality.

**Questions:**

See weakness.

---

> ### Author Response · Authors · 2025-11-20
> **Response to the official review**
>
> Thank you for reviewing our paper.
>
> ## Necessity of token limitation through logit modifiers
>
> > why enforcing token limits is necessary
>
> As we briefly mentioned in Line 041, LLMs sometimes generate endless output, and it is unforeseeable and practically unmanageable solely through prompts. LLMs don’t recognize the number of their output tokens and we empirically demonstrate that prompts like “keep the output as compact as possible” aren’t strictly followed (See Appendix B.11 Table 4 for the experiments on such prompts). For chat-based applications, forced termination is an option because users rarely care about truncated outputs. However, in scenarios of agent-based applications where autonomous agents communicate each other using structured text without human intervention, such termination causes parse errors that subsequently break downstream processes. Our motivation is to terminate the generation with preserving grammatical correctness.
>
> > does it improve overall accuracy or just ensure early termination?
>
> When we use TruncProof solely, it ensures early termination but the overall accuracy, i.e. semantic correctness, is not improved. However, by incorporating advanced strategies such as Beam Search and Monte Carlo Tree Search (as described in Section 4.2), the proposed approach achieves both early termination and improved semantic correctness.
>
> This is because TruncProof can help LLMs to explore more plausible (i.e., lower perplexity) outputs for semantic correctness.  We would like to emphasize that existing methods do not obtain significant benefits from these strategies, as experimentally demonstrated in Line 410 and the rows of BS and MCTS in Table 1.
>
> **Question**:
>
> > simply applying standard grammar-constrained decoding (e.g., CFG-based methods) with an explicit token
>
> We would appreciate it if you could also point us to methods based on explicit tokens. We are considering adding experiments that compare our approach with these methods.
>
> ## step-by-step walkthrough of each phase of TruncProof
>
> Thank you for your suggestion. Figure 2(c) only showed the state at a single point in runtime, so we will include a more comprehensive diagram of our algorithm to our revised manuscript.
>
> ## preprocessing / runtime overhead
>
> > the overhead introduced by TruncProof, both in preprocessing and runtime, especially whether scaling up grammar size will affect decoding efficiency, should be quantified and discussed.
>
> We report the preprocessing time for JSON (1 min) and C (5 min) in Line 353.
> Runtime overhead in each method can be identified as the decrease in tokens/sec from "No constraint” in Table 1 (e.g. 45.87 -> 38.93 for Gemma2-2B).
>
> Since the comparison of this decrease requires a bit of extra effort, we will explicitly report the overhead in Table 1 in the revised manuscript.
>
> Both in preprocessing and runtime, the overhead is in theory proportional to the number of accept sequences $\mathcal{A}$ described in Line 237, whose upper bound is $O(|\Sigma_T|^2)$.
>
> ## metrics for semantic correctness or downstream accuracy
>
> > the evaluation focuses primarily on syntactic validity rather than overall task performance; adding metrics for semantic correctness or downstream accuracy (e.g., in code completion, data-to-JSON generation, or semantic parsing) would help clarify whether the proposed constraint mechanism provides real benefits beyond ensuring grammaticality.
>
> For JSON tasks, we consider Exact-Match (It checks whether Python parses the string into the same JSON object as the ground truth. Defined in Line 336) is the metric that represents  semantic correctness.  We demonstrate that our TruncProof improves this metric (Line 405), and we think that the real benefits of TruncProof are demonstrated in the use-case of JSON generation beyond just ensuring grammar correctness.

---

### Author Response · Authors · 2025-12-02
**Comments to all reviewers and meta-reviewers (1/2)**

To all reviewers, meta reviewers

We sincerely appreciate your constructive feedback and insightful questions.
Based on your comments, we have revised the manuscript. All changes are highlighted in blue.
Below is a summary of the revisions together with the corresponding reviewer comments:

* **Clarification of the motivation for TruncProof (Line 046-048).** Reviewer V4hk and wgCt requested a clearer explanation of the motivation behind TruncProof (i.e., the importance of Grammar-Constrained Generation under token limitations). In response, we have expanded the discussion in Lines 046–048 to clarify this point, emphasizing that output truncation leads to severe issues in agent-based applications.

* **Additional related works (Section 3).** Reviewer AGnR pointed out that two recent works i.e., LLGuidance and GreatGramma were not cited. In the revised manuscript, we have added both to the Related Works section (§3). Note that we do not include them in our experimental comparison because (i) as discussed in §3, neither method is capable of ensuring grammatically valid outputs within a predefined token limit, which is the primary challenge addressed by TruncProof, and (ii) the implementation of GreatGramma is not publicly available. We believe that our results (Section 5.2, Table 1, and Figure 4(b)) sufficiently demonstrate the superiority of our method in terms of robustness under strict token limitations.

* **Diagrams of TruncProof (Figure 2 & 3).** Reviewer V4hk suggested providing a clearer walkthrough of TruncProof to make the method understandable at a glance. In response, we have added two figures in the revised manuscript: Figure 2 presents an overview of TruncProof with a caption that briefly summarizes the workflow, and Figure 3 provides a graphical explanation of Equation 5. In addition, for improved clarity, we have reorganized the explanation in §4.1 by first describing the runtime phase in §4.1.1, followed by the precomputation phase in §4.1.2.

* **Improved clarification of Table 1 (Table 1).** Reviewers V4hk and 64zc raised concerns regarding the runtime overhead introduced by constrained generation. Although the initial manuscript already reported tokens/sec for each method and each decoding strategy in Table 1 and it allows the overhead to be computed as [tokens/sec of the method] − [tokens/sec of the No Constraint setting], we agree that this required additional effort for the reader. Therefore, in the revised Table 1, we have explicitly added the overhead values in the final column for improved clarity.

* **Clarification of the languages and formats covered by LL(1) grammars (Line 350-354).** Reviewer AGnR suggested clarifying which languages can and cannot be defined using LL(1) grammars. In the revised manuscript, we have added examples of languages that fall within and outside the scope of LL(1) when introducing our experimental settings (Lines 350-354). Because TruncProof assumes an LL(1) grammar, languages that cannot be defined using LL(1) (e.g., SQL, Python, Java, or Go, which are all mentioned by wgCt) are excluded from our experiments, even if they are considered in existing literature.

---

### Author Response · Authors · 2025-12-02
**comments to all reviewers and meta-reviewers (2/2)**

We also note that some concerns and questions raised by the reviewers had already been addressed / resolved in our initial submission:

* Reviewer V4hk inquired about the preprocessing overhead with respect to grammar size. We reported the processing time for JSON (specified in Appendix B3) and a subset of C (specified in Appendix B9) in the first manuscript (Line 391 in the revised version).
* Reviewer 64zc and AGnR raised questions regarding the behavior of each constraint method under different token limitations. This was already addressed in Appendix B10 of the initial manuscript. As Figure 7 shows, TruncProof consistently outperforms SynCode, and the performance gap becomes particularly large under strict token limits.
* Reviewer wgCt and AGnR raised concerns regarding the limited coverage of LL(1) grammars. Although it is true that some languages cannot be defined using LL(1), we would like to emphasize that JSON (Appendix B3) and the subset of C (Appendix B9), both of which are covered by LL(1), are widely used formats and programming languages, and supporting them brings substantial practical benefits (Line 130). We believe that leveraging the underlying LL(1) property (L123-127) is a reasonable and practical choice for achieving the exclusive advantage of TruncProof, namely the ability to generate grammatically correct outputs within a given token limit.

* Reviewer wgCt raised concerns regarding the coverage of our experiments. However, the text-to-JSON benchmark we used, JSON-Mode-Eval, is widely adopted by existing methods (SynCode, XGrammar), and we believe that we successfully demonstrate the superiority of our TruncProof in our research scope. In addition, we think that an example of text-to-C generation implies TruncProof's potential to perform more flexible token truncation compared to simply reducing whitespace (Line458-463).

We believe that the above revisions, together with the information that was already included in the initial manuscript, sufficiently address all concerns raised by the reviewers.

---

### Meta-Review · Area_Chair_t5c5 · 2026-01-05

**Summary:**

**Summary**：
The paper proposes "TruncProof," a grammar-constrained generation framework that enforces strict token limits on LLM outputs. By leveraging the lookahead properties of LL(1) parsers, the system dynamically estimates the minimum tokens required to complete a valid grammatical structure and masks out tokens that would cause the generation to exceed the budget. The primary goal is to prevent "crash-on-truncation" errors (e.g., unclosed JSON brackets) which disrupt downstream agentic workflows.
**Rebuttal Conclusion**：
Despite the authors' successful demonstration of the method's robustness in the specific domain of JSON generation for agents, the general agreement leans towards rejection due to the fundamental limitations of the chosen technical approach. While Reviewer 64zc championed the work for its intuitive solution to a practical problem, Reviewers wgCt and AGnR raised critical concerns regarding the regressiveness of the methodology. The reliance on LL(1) grammars restricts the framework from supporting mainstream programming languages (Python, SQL, Go), which are supported by existing SOTA methods (e.g., SynCode, XGrammar). Consequently, the paper is viewed as a specialized engineering patch for JSON agents rather than a generalizable contribution to constrained decoding suitable for ICLR.

**Reviewer Concerns:**

**Concerns Addressed by the Rebuttal**
•**Motivation and Prompting Baselines**: Resolved by adding experiments (Appendix B.11) comparing TruncProof against prompt engineering (e.g., "keep output compact"). The results showed that prompting fails to stick to strict limits (4%-16% accuracy) while TruncProof achieves 100%, justifying the need for algorithmic constraints.
•**Overhead and Efficiency**: Resolved by quantifying the costs. The authors clarified that precomputation is a one-time cost (1 min for JSON) and runtime overhead is negligible (~3.9ms/token), addressing concerns from Reviewers V4hk and AGnR.
•**Sensitivity Analysis**: Addressed by providing a detailed trade-off analysis (Figure 7, Appendix B.10) across various token budget ratios ($e=1.0$ to $1.5$), demonstrating consistent robustness compared to baselines.
**Outstanding Concerns**:
•**Technological Regression (LL(1) Limitation)**: This is the primary ground for rejection. Reviewers wgCt and AGnR correctly pointed out that relying on LL(1) grammars is a theoretical step backward compared to prior works that support general Context-Free Grammars (CFGs). The method inherently cannot support complex languages like Python or SQL, severely limiting its scientific contribution.
•**Limited Experimental Scope**: The evaluation is heavily over-fitted to JSON generation. The code generation experiment relies on a trivial "Subset of C" and lacks comprehensive benchmarks (e.g., HumanEval). The authors admitted this limitation but could not address it due to the inherent constraints of the LL(1) parser .
•**Theoretical Approximation**: Reviewer AGnR noted that the token counting mechanism involves over-approximation (e.g., tokens spanning multiple grammatical units). While empirically effective for simple JSON, the theoretical derivation lacks the precision expected for a top-tier conference.

**Reviewer Scores:**

I think the estimated final opinions are split: 8, 4, 4, 4.
Reviewer 64zc (Score: 8) remained a strong supporter, valuing the practical robustness in JSON tasks. However, Reviewers wgCt and AGnR (Score: 4) maintained their objections. They argued that while the method solves a specific engineering pain point (agent crashes), the solution is too narrow. Reviewer V4hk (Score: 4) acknowledged the improved motivation but remained unenthusiastic regarding the limited scope.

---

### Decision · Program_Chairs · 2026-01-26

Reject